# Alternative Methods of Sterilization in Dental Practices Against COVID-19

**DOI:** 10.3390/ijerph17165736

**Published:** 2020-08-08

**Authors:** Enzo Cumbo, Giuseppe Gallina, Pietro Messina, Giuseppe Alessandro Scardina

**Affiliations:** Department of Surgical Oncological and Stomatological Disciplines, University of Palermo, 90127 Palermo, Italy; enzo.cumbo@unipa.it (E.C.); giuseppe.gallina@unipa.it (G.G.); pietro.messina01@unipa.it (P.M.)

**Keywords:** COVID-19, SARS-CoV-2, bioaerosols, sterilization, dentistry

## Abstract

SARS-CoV-2, and several other microorganisms, may be present in nasopharyngeal and salivary secretions in patients treated in dental practices, so an appropriate clinical behavior is required in order to avoid the dangerous spread of infections. COVID-19 could also be spread when patients touches a contaminated surface with infected droplets and then touch their nose, mouth, or eyes. It is time to consider a dental practice quite similar to a hospital surgery room, where particular attention should be addressed to problems related to the spreading of infections due to air and surface contamination. The effectiveness of conventional cleaning and disinfection procedures may be limited by several factors; first of all, human operator dependence seems to be the weak aspect of all procedures. The improvement of these conventional methods requires the modification of human behavior, which is difficult to achieve and sustain. As alternative sterilization methods, there are some that do not depend on the operator, because they are based on devices that perform the entire procedure on their own, with minimal human intervention. In conclusion, continued efforts to improve the traditional manual disinfection of surfaces are needed, so dentists should consider combining the use of proper disinfectants and no-touch decontamination technologies to improve sterilization procedures.

## 1. Introduction

A dental office is an environment in which bioaerosols are regularly present, which are generated especially during the use of ultrasound or other hand pieces that produce sprays. The production of these microparticles, created mostly inside the patient’s mouth, are therefore microbially contaminated and seem an inevitable phenomenon, because they are closely linked to the activity of the dentist. It is known that if patients carry microbes in the mouth and respiratory tract, they can spread them into the air by aerosolization [1,2].

As a result, dentists can easily be exposed to infections due to the short distance between them and their patients, who could also transmit microbes by sneezing, coughing, or simply speaking. Paradoxically, if a patient tries to reduce the spread of his disease by coughing on the hand, as a consequence, he produces more aerosols of small particles that are potentially more suitable for diffusion in the air.

There are also other conditions to consider, such as the reduced spaces of dental operating rooms and, consequently, the sharing of a small space and the breathed air that could easily lead to the transmission of biological agents. An adult man breathes about 700 L of air per hour, and, in the event of hyperventilation (which can occur if the patient is anxious or frightened due to dental treatment), the total amount of air breathed per hour can significantly increase; the more air you breathe, the greater the risk of a microbiological transmission.

Life-threatening viruses, such as SARS-CoV-2, responsible for a severe acute respiratory syndrome that, according to WHO, has a mortality rate of 3.4%, could be present in the respiratory tract [3,4].

In dental offices, any surface can be contaminated with viruses through contact with infectious body fluids or through the sedimentation of airborne viral particles [5].

On those infected surfaces, in order to represent a source of infectious risk, the virus involved must be able to survive until it encounters a new sensitive host; in fact, viruses, in general, are the so-called “obligate parasites”, because they cannot multiply or propagate outside specific host cells.

In order to effectively fight viral infections, it is essential to know the behavior of viruses in different environmental conditions in order to improve countermeasures, making them effective especially when the infection in question assumes particular importance because it is able to put patients’ lives at risk.

Although, in the field of medicine, doctors are well-prepared on how to fight virus-based diseases, a new enemy like SARS-CoV-2, also known as COVID-19, can test the entire healthcare system.

This microbe belongs to a family of single-stranded RNA viruses (well known as Coronaviridae) that are said to be zoonotic or transmitted from animals to humans.

At present, there is no effective medicine for SARS-CoV-2, so the only drugs used are those targeted for side effects in infected patients. In this case, the best defense weapon against the spread of the virus seems to be the attempt to limit the passage from an infected patient to a new host in order to minimize the exponential spread.

CoV-19. To transmit the infection to a sensitive host, it must be able to survive the aerosol process and persist in the air for a long enough time to allow transfer from one person to another.

Numerous factors influence the survival of viruses in the air: (1) particle size, (2) atmospheric temperature and relative humidity (RH), (3) nature and composition of the aerosol, (4) atmospheric gases, and (5) irradiation [6].

All the factors mentioned are involved differently in promoting or delaying the survival of pathogenic viruses in aerosols, which are substantially airborne particles of various sizes.

### 1.1. Size of Particles

Particles behave differently depending on their size: the larger ones settle quickly, but smaller-sized particles can remain suspended in the air for a long period of time.

In a room with calm air, particles with a diameter of only 10 nm that fall through a height of 2 m take about 12 min to stabilize; under the same conditions, particles with a diameter of 40 nm would only take 40 s.

Furthermore, in the presence of turbulence or a simple flow of air, particles can move very far from the point of their generation, or they can remain suspended for several minutes.

The smaller particles of an aerosol have a greater transmission potential for infections, because they can better penetrate and settle in the smaller passages of human lungs.

The biggest threat of airborne infection in dentistry is known to be from particles smaller than 50 μm due to their ability to stay airborne and the potential to enter the respiratory tract [7].

The largest particles, greater than 50 microns in diameter, are called “splatters” and behave in a ballistic way; in fact, these droplets are expelled like bullets in a trajectory initially quite straight that, secondly, becomes curved. Finally, they contact a surface or fall to the floor. These particles are too large and heavy to remain suspended in the air and are suspended in the air only for a short time.

However, if the droplet starts to evaporate, its size becomes smaller and smaller and acquires the potential to remain in flight for a longer period of time. Therefore, splatter droplets can be considered a potential source of infection in a dental environment; in fact, they have been implicated in the transmission of various diseases, such as SARS and herpes.

### 1.2. Atmospheric Temperature and RH

Several studies show that the evaporation of water from small aerosol particles depends on the atmospheric temperature and the relative humidity level (RH).

The evaporation of water leaves behind a residual particle that can contain organic and inorganic materials, as well as biological agents that, if not damaged by the drying process, are potentially infectious for sensitive guests [8].

Therefore, the problem related to the resuspension in the air of previously dried infectious materials that could remain on work surfaces is of fundamental importance; in fact, if the infectious agents manage to survive, the direct or indirect contact of sensitive hosts with these surfaces could lead to the spread of infections.

Aerosol humidification studies have shown an increase in the recovery of infectious particles from certain types of viruses, such as the flu virus, indicating that the simple rehydration of virus particles in the air leads to their reactivation [9].

In contrast, other studies have shown that humidification can reduce the recovery efficiency of some types of viruses [10].

In any case, temperature and RH are the two most important factors, which often act in combination, in determining how long viruses survive in the air state. In general, the ability of viruses to survive in the air state is inversely proportional to air temperature, but there are exceptions.

Numerous studies on the infectivity of viral aerosols provide clues to understand how different climates influence the onset of many viral diseases. Meteorological parameters are in fact important factors influencing infectious diseases, such as severe acute respiratory syndrome (SARS) and influenza.

Yueling et al. showed a significant negative association between COVID-19 mortality and room temperature, as well as absolute humidity. Indeed, the daily mortality of COVID-19 appears to be positively associated with the daytime temperature range (DTR) but negatively with the absolute humidity. Hence, temperature and humidity are factors influencing COVID-19 mortality [11].

It is generally believed that lipid-containing viruses survive better at low RH levels, and high RH levels are more conducive to aerial survival than lipid-free viruses [12,13].

Under certain experimental conditions, some types of viruses were found capable of surviving well at high and low RH levels but were sensitive to inactivation in the medium RH range; an example of such viruses is the flu virus [9].

Contrary to this, other studies of the human coronavirus have shown that they survive the air state better at 20 °C when the relative humidity is maintained at 50%.

Other studies conducted on the human coronavirus have shown that, when the aerosols were kept at 20 °C/80% relative humidity, its half-life was only 3 h.

Reducing the air temperature to 6 °C/80% relative humidity can cause an increase in the half-life of the coronavirus to almost 87 h [14].

The ability of low temperatures to overcome the effect of RH on a wrapped virus such as coronavirus suggests that a reduced fluidity of the lipid bilayer may be involved in limiting the access of inactivating factors to the virus’ nucleic acid or protein components.

### 1.3. Nature and Composition of the Spray and Collection Fluids

It is known that the stability of an aerosol in the air is influenced by the composition of the fluid from which the virus is aerosolized [15].

Some studies on viral aerosols have been conducted using artificial spray fluids, and other studies have used natural substances such as saliva. Consequently, the decay rates of artificially generated viral aerosols, under laboratory conditions, can be different from the natural aerosols of the same virus derived from body secretions, and in general, a protective effect of the natural spray is recognized.

In dentistry, bioaerosols can be considered a mix of natural secretions, such as saliva mixed with blood, microorganisms, mucous membrane cells, restoration materials, dental particles, and water from handpieces normally used in dental practices. Such aerosols are very common during different treatments, such as scaling and root planning using ultrasonic scalers or air polishing procedures; washing or drying with air-water syringes; and preparing the teeth with rotating instruments such as high-speed dental turbines, handpiece micromotors, or air abrasion [16,17].

### 1.4. Atmospheric Gases

All experimental studies in this area have attempted to discover and develop chemicals in order to prevent the transmission of viruses by air.

The viruses present in aerosols can be inactivated by the action of chemical gases such as propylene glycol vapors, which, for example, are effective against flu virus aerosols [18].

Other studies have been conducted on hydrogen peroxide, chloramine, and hexylresorcinol tested on various viruses present in the air. The minimum concentrations of these compounds, necessary to determine a 99.9% reduction in 30 min, varied from 5 to 20 mg/m3; however, much higher concentrations of the same disinfectants were needed to bring about a 99.9% reduction in the virus titer on a contaminated surface.

In order to improve the effectiveness of these chemical gases, their use has been proposed in combination with controlled RH levels; this combination could offer prospects for the effective disinfection of recycled air.

### 1.5. Irradiation

Both ionizing and nonionizing electromagnetic radiation affect the biological activities of microorganisms such as viruses depending on the wavelength, and their effects on biological materials differ significantly.

Radiation energy is also absorbed by materials around the microorganism, and these phenomena can indirectly influence the virus; in fact, radiation could alter, first of all, this material that becomes harmful to the virus due to the absorption of radiated energy that can be secondarily transferred to the virus, damaging it.

UV radiation can introduce changes in bioaerosols, but its composition can influence what happens to the virus. Although various changes in proteins and nucleic acids are known to be caused by radiation, it is important to evaluate whether they are relevant to the loss of the biological activity of viruses.

Among the ionizing radiations, we must also mention gamma rays, widely used in the field of sterilization, which are electromagnetic radiation derived from the radioactive decay of atomic nuclei.

Gamma rays have electromagnetic waves of shorter wavelengths and, consequently, transmit the maximum energy of the photon, with an enormous ability to penetrate and kill living organisms; in fact, among their applications, there is the sterilization of medical equipment [19].

## 2. Alternative Methods of Sterilization

Dentists could treat, during daily practice, asymptomatic patients unaware of having been infected with SARS-CoV-2, which, in these cases, could be dangerously present in their nasopharyngeal and salivary secretions; therefore, an appropriate clinical behavior is needed in order to avoid the uncontrolled spread of the infection [20,21].

COVID-19 could also spread when dental patients touch a surface contaminated with infected droplets and then touch their nose, mouth, or eyes [22]; if proper precautions are not taken, the dental office can expose patients to cross-infections.

There are two different ways to deal with the problem: the first is to identify infected patients and postpone treatments (if possible) or refer them to the appropriate hospitals; the second is to consider all patients highly dangerous because they are potentially infected.

It is time to consider a dental practice quite similar to a hospital surgery room, where particular attention should be paid to problems related to the spread of infections caused by air and surface contaminations, especially a time when viruses such as SARS-CoV-2 have emerged as an important public health problem due to their ability to spread through close person-to-person contact.

There are so many aspects to focus on in order to reduce the spread of this dangerous viral infection and include environmental treatments, air ventilation, the use of personal protection, etc.

In dentistry, conventional cleaning and disinfection have been used for several years, but their effectiveness can be limited by several factors—first and foremost, the right choice of products and the procedure adopted [23,24,25,26].

Serious errors such as the use of incorrect chemicals, inadequate dilutions, inadequate contact times, inadequate or numerically inadequate microfiber cloths or paper towels, and incorrect application methods that can spread pathogens from one surface to another can also occur.

Therefore, the dependence of the human operator appears to be the weak aspect of all procedures.

The improvement of these conventional methods requires the modification of human behavior, which is difficult to achieve and sustain [27,28,29].

As alternative sterilization methods, there are some that do not depend on the operator, because they rely on devices that perform the whole procedure on their own, with minimal human intervention. These methods, which are often called noncontact disinfection systems (NTD), can be applied in the field of dentistry, especially now that important sterilization problems have arisen due to COVID-19.

### 2.1. Ozone

Ozone (O_3_), also known as trioxygen, is an inorganic gaseous molecule; under standard conditions, its color is pale blue, and its presence is characterized by a pungent odor reminiscent of chlorine; most people hear it at concentrations of just 0.1 ppm in the air.

O_3_ as an oxygen allotrope that is much less stable than the diatomic allotrope O_2_; in fact, it breaks down into oxygen, with a half-life of about 20 min.

Ozone is produced from O_2_ through ultraviolet light or atmospheric electrical discharges and is present in very low concentrations, with its maximum concentration in the so-called “ozone layer” of the stratosphere, which absorbs most of the sun’s ultraviolet radiation.

Ozone is used commercially only at low concentrations, so any concern about its instability and the risk that both concentrated gas and liquid ozone may explosively decompose at high temperatures appears insignificant [30].

The antiviral and antimicrobial properties of O_3_ have been well-documented, and several macromolecular targets may be involved; this gas has been shown to kill the SARS virus, which structure is quite similar to the new SARS-CoV-2.

More precisely, ozone destroys viruses by spreading through the protein coating in the nucleic acid nucleus, causing damage to viral RNA. At higher concentrations, ozone destroys the capsid or the outer protein shell by oxidation.

Most research efforts on the viricidal effects of ozone have focused on the propensity of ozone to break down lipid molecules in multiple-bond configuration sites. In fact, once the lipid envelope of the virus is fragmented, its DNA or RNA nucleus cannot survive.

Wrapped viruses, such as SARS-CoV-2, are generally more sensitive to physicochemical challenges than naked virions. Although the effects of ozone on unsaturated lipids are one of its best documented biochemical actions, ozone is known to interact with proteins and carbohydrates.

Unlike liquid sprays and aerosols, gaseous ozone can easily penetrate all areas within a room, including cracks, fixtures, fabrics, under furniture surfaces, and on the floor [31].

Ozone, with its great oxidizing power, therefore has many applications in the field of medical sterilization, especially when it is necessary to sterilize different types of surfaces (smooth or porous) containing dry or wet films of different viruses in the presence and absence of cellular debris and biological fluids [32].

Such conditions are substantially present in any dental practice; therefore, its use could be widely adopted in these environments, especially in the case of the pandemic spread of dangerous viruses such as COVID-19.

In this regard, some studies have shown that ozone gas is able to effectively kill the viruses transmitted by aerosols, with a reduction in the presence of viruses up to 99% [31,33].

Certainly, its use requires some precautions, because its high oxidizing potential makes ozone a powerful respiratory and polluting hazard; its presence, above concentrations of about 0.1 ppm, can cause damage to mucous and respiratory tissues in humans and, also, to plant tissues [34].

If the dental office has more than one operating room, this toxicity problem can be easily solved by using these rooms alternately; when a room is without people inside, the ozonator can be switched on without any risk. Furthermore, to speed up the disinfection procedure, ozone can be converted into oxygen fairly quickly by means of a catalyst, and, in the absence of the latter, as an alternative method, strong ventilation is suggested.

It is also important to emphasize that, when an ozone generator is used, some other precautions should be taken, such as the use of the remote control on the device or a timer, in order to always remain outside the room during the sterilization process; the procedure must also be performed with closed windows.

Other significant disadvantages are its ability to corrode and ruin certain materials, such as natural rubber, especially in the event of prolonged exposure; therefore, these materials can be temporarily removed if necessary [35].

During environmental sterilization by ozone, there is a correlation between the RH and efficiency of this procedure; in fact, it is assumed that the maximum enhancement effect is obtained by first increasing the ozone to the maximum level, followed by a burst of water vapor for increased RH greater than 70%—preferably, > 90%. Otherwise, at ambient RH, the degree of inactivation is lower and more variable; therefore, the concentration of ozone, RH, and the exposure time seem to be fundamental.

There are other important considerations that have emerged from studies in the literature—for example, both dry and wet virus films were found to be equally sensitive to ozone treatment and, at the same time, the nature of the surface on which the viruses are located, not that it makes any difference. The latter aspect can be considered positively, because the surfaces of fabric, plastic, metal, and glass are equally sterilizable, even when cellular debris, including blood, is present [36].

Comparative studies have shown results on different sterilization procedures for operating theaters, and, in conclusion, the effectiveness of ozone seems to be comparable to the use of UV radiation or 2% glutaraldehyde, but without wasting products, ozone is also easy to insulate and does not require washing [37].

### 2.2. Air Ionization

An air ionizer is a device which, through high voltage, can generate negative ions, which are particles with extra electrons, which give a net negative charge to the particle; conversely, the positive ions lack the electrons for which they have a positive net charge.

Ionizers use metal surfaces charged with electricity to create ions from air or electrically charged gases that attach to airborne particles that are then electrostatically attracted to a charged collector plate [38].

The simpler ionizer scheme contains a row of wires and a stack of large flat metal plates; between those wires and plates, a negative voltage of several thousand volts is applied.

For safety reasons, the collector plates have a very low current (<80 μA); however, a high voltage ionizer can produce several billion electrons per second.

The air flow first flows through the spaces between the wires and, then, passes through the stack of plates. Thanks to the high voltage, an electric corona discharge ionizes the air near the electrodes, which ionizes the particles in the air flow, which are diverted to the grounded plates due to the electrostatic force; finally, the air flow removes the particles accumulated on the plates [39].

Ionizers, which have been used to eliminate or reduce both bacterial and viral infectious agents suspended in the air [40,41], can be divided into fanless ionizers and fan ionizers.

Fanless ionizers are generally smaller and quieter devices that are less efficient in air purification; fan ionizers clean and distribute air much faster.

Some of these devices are called generations of wind electrons (EWG), which are filtration systems capable of purifying the ambient air from bioaerosols whose presence can be dangerous, for example, in operating rooms.

Usually, they are small devices capable of generating continuous air circulation through a network of electrodes; then, the air flow, which is drawn into the device, is sterilized by the electric field.

Numerous studies have been conducted over the years to develop different air purification procedures.

Among the various air sterilization technologies, filtration methods have become very popular [42].

The EWG air treatment significantly changes the characteristics of the microbes present in the air and shifts the main peak of the dimensional distribution of the microorganisms present in the air in the coarser bioparticles.

The use of EWG sterilization seems to considerably reduce the concentration of live microbes present in rooms; in fact, several studies have shown that sterilization using the EWG system seems to be very promising, because the microbial load inside the chamber is greatly reduced thanks to the high voltage field (from 5 Kvolt to 15 Kvolt), which causes irreversible damage to the cellular film and disturbances in the replication of microbes following breakages of double-stranded DNA [43].

However, the effectiveness of the sterilization process depends on several factors—for example, the size of the room and the number of people inside it, the working time of the device, and, consequently, the volume of air treated per hour.

The best results are reported in confined spaces (e.g., 30 m^3^) with no more than three people.

Due to the high voltage, significant ozone emissions have been demonstrated during EWG air filtration; it is known that the ozone emission can further reduce the number of live microorganisms in the air; however, the increase in the concentration of ozone in the rooms adversely affects the health of people.

Fortunately, studies have shown that applying a carbon filter to the EWG device results in a massive reduction in ozone emissions; of course, the addition of these special carbon filters also helps to collect biological and nonbiological aerosol particles [17].

Numerous studies have shown a significant reduction in the spread of viral infections also among animals, and these results represent the hope of further applications in the human medical field [44,45].

The electronic wind generator system allows people to stay in the operating room while it is turned on, with the advantage of not interrupting the workflow. All these devices are portable, so that they can be moved from one operating room to another to optimize time; obviously, this system does not work on contaminated surfaces but only against air contamination; therefore, it must be integrated with other methods, such as those based on chemical products.

### 2.3. Photocatalytic Oxidation

Photocatalytic oxidation (PCO) is a technology in the heating, ventilation, and air conditioning (HVAC) sector.

The main function of typical HVAC systems is to control the temperature and relative humidity of the ambient air; moreover, thanks to the presence of mechanical or electrostatic filters, it is also possible to remove polluting air particles. If a carbon filter is present in the system, the gaseous contaminants or vapors emitted are eliminated.

However, it is well-proven that filters, used to remove particles from the air, can pollute the air instead of cleaning it, especially if the humidity is high.

In the case of absorbent filters, if the temperature or humidity increases, the volatile organic compounds (VOCs) can be desorbed instead of being absorbed by contaminating the air [46,47].

Instead of adsorbing VOCs on the absorbent filter, the photocatalytic process is able to oxidize them in CO_2_ and H_2_O and control biological contaminations.

It is important to specify that PCO, which can use UV radiation to energize the catalyst (usually TiO_2_) and oxidize bacteria and viruses, is not a filtering technology, as it does not trap or remove particles but is sometimes simply coupled with technologies filtering for air purification; in fact, it can be mounted on an existing forced air HVAC system.

The effectiveness of the photocatalytic filter depends on several parameters, such as the total air change speed, the type of filter, and the relative humidity; in any case, further research is necessary to establish whether this system is valid as a control of biological contamination in the field of dentistry [48,49].

### 2.4. H_2_O_2_-Based Systems

#### 2.4.1. Aerosolized Hydrogen Peroxide (AHP)

The decontamination system based on hydrogen peroxide in aerosols is a “no-touch” method that uses hydrogen peroxide (3% to 7%), which can be combined with the addition of silver ions (<50 ppm).

This method, also called “hydrogen peroxide in dry mist”, is based on the injection of an aerosol with particles ranging from 2 to 12 μ in size; this first phase is followed by passive aeration.

The results obtained from these procedures are controversial; some studies have shown a significant reduction in microbes, including spores, but other researchers have shown incomplete eradication.

Like many other infection control strategies, there are currently no randomized controlled trials on the effectiveness of these systems in preventing healthcare-associated infections [50,51,52].

#### 2.4.2. H_2_O_2_ Vapor

In this system, a heat generator is involved to create, from H2O2, a high-speed air/steam flow (30–35%).

These generators are remotely controlled and may be able to measure the concentration of hydrogen peroxide vapor; some systems also have an integrated ventilation unit and dehumidifier designed to reach a humidity level set before the cycle starts [53,54].

Studies of these procedures have shown some efficacy against a variety of pathogens, including viruses and prions [55,56].

#### 2.4.3. UV Rays

Ultraviolet (UV) light is an electromagnetic radiation invisible to the human eye, since its wavelength is shorter than the visible one and is between 400 nm and 100 nm, although, in some conditions, young people may be able to see ultraviolet light up to wavelengths of approximately 310 nm [57].

The UV spectrum, as a function of frequency, is divided into five segments: UV under vacuum (40–190 nm), UV far (190–220 nm), UVC (220–290 nm), UVB (290–320 nm), and UVA (320–400 nm) [58].

Although UV rays are present in nature because the sun is a primary natural source, there are also artificial sources such as curing lamps, tanning booths, germicidal lamps, black lights, halogen lamps, mercury vapor lamps, discharge lamps high intensity, fluorescent and incandescent sources, and some types of lasers.

Due to its ability to provoke chemical reactions and to excite fluorescence in materials, UV light has a large number of applications in various fields, including medicine.

In dental diagnostics, UV lights are used for the fluorescence of teeth with radiation exposure in small areas and doses not exceeding 5 J/cm2 [59].

UV rays are also one of the oldest known methods for decontamination from viruses, bacteria, and fungi; UVC germicidal lamps (220–290 nm) are mainly used in sterilization procedures, but the optimal wavelength for the best results is about 253.7 nm, since the maximum absorption wavelength of a molecule of DNA is 260 nm.

After UV irradiation, the DNA sequence of microorganisms can form pyrimidine dimers, which can interfere with DNA duplication, as well as lead to the destruction of nucleic acids and make viruses noninfectious [60]. In addition, the viral nucleic acid type, the host cell repair mechanisms, and the capsid structure of the virus play an important role in virus inactivation. UV radiation restructures the nucleic acid of the germs and destroys its replication ability; this is the reason why the viral nucleic acid type can play a critical role in the inactivation of the virus by UV rays. More precisely, viruses with RNA or DNA may be less sensitive to UV rays; important seems to be the presence or absence of the cell wall and its thickness.

These are the reasons why ultraviolet radiation can be used in the field of sterilization—in particular, in environmental control against air and surface contaminations.

UV irradiation seems to be effective even on surgical instruments, but if they have overlapping parts that remain in the shade, these procedures are not recommended; as a result, UV-C rays are mainly used in air and water purifications with good results; therefore, research on UV disinfection continues today [61].

Ultraviolet light has proven effective against corona viruses and, therefore, could be used against COVID-19 both in the case of bioaerosols and in the sterilization of contaminated environmental surfaces in which this microorganism is present—in particular, on products of unstable composition that cannot be treated by conventional means [62,63].

Contrary to most chemical disinfectants, UV rays have been well-recognized as an effective method for inactivating microorganisms, but their effectiveness for inactivating germs has been related to several parameters, such as the level of irradiation, the duration of irradiation (in general, the UV dose for a 99% viral reduction is two times higher than for a 90% viral reduction), and RH; at a high RH, a higher UV dose is required to inactivate viruses on contaminated surfaces [64].

The intensity of the UV light is dissipated with the square of the distance from the source, and this unfortunately limits the ability of the individual UV-C devices to disinfect large areas. A UV lamp intensity of 40 μW/cm^2^ in the center of the work area is recommended to ensure surface decontamination; however, there are several sources that provide a list of UV dosages needed to kill a wide range of microorganisms [65].

The doses of lethal UV radiation necessary for viruses in the air are lower than those of viruses on surfaces; this can be explained by the fact that viruses can form aggregations on surfaces and that viruses can be less sensitive to UV rays if associated with water [66,67].

It is known that all pathogens are sensitive to UV radiation, but this susceptibility between them is different. To classify this vulnerability to UV light, there is an index called Z; the higher the index, the more vulnerable the microorganism to UV exposure [68].

In dentistry, there are two different methods of achieving a significant reduction of microorganisms by UV; the first method is based on devices capable of being effective only on microbes present in the air. In order to improve the quality of the ambient air, the air is forced into the device where the UV source is present, and, after adequate sterilization, it is released into the environment. This procedure allows people to stay in the room when the device is turned on, because there is no exposure to dangerous radiation, so air sterilization can be done for several hours without any risk. On the other hand, these devices are not effective on surfaces, including the floor, where microbes can be widely present.

The second method is completely different, because it is based on the exposure to UV rays of the entire environment through the use of fixed UV-C sources strategically positioned to expose the largest possible area. Alternatively, mobile supports can be used for a better and more flexible orientation of UV light towards contaminated surfaces.

When considering UV-C irradiation to inactivate viruses on surfaces, special attention must be paid; in fact, the growth of microorganisms may occur in shaded areas, such as cracks or crevices where UV radiation may not arrive. The use of a mobile UV sterilization system could be the way to overcome this problem, even if the procedure becomes operator-dependent.

This method, which exposes the entire room to UV rays, has the advantage of being effective simultaneously on surfaces and in the ambient air but cannot be performed if people are in the area due to negative health effects, such as the risk of skin erythema and photokeratitis.

Fortunately, most dental offices have two operating rooms; while one is in use, the other can be treated with UV radiation.

All UV systems have numerous advantages; first of all, they are easy to use, and, unlike chemical gases such as ozone, they do not require the sealing of doors or air intakes; another advantage is that the whole procedure is relatively short.

Indeed, depending on the reflective surfaces in the room, an effective cycle can last differently, becoming shorter and more efficient if reflective objects and surfaces are widely present in the irradiated area.

Another side effect with the use of UV rays is the damage or discoloration of surfaces, especially if they are made of plastic [69].

### 2.5. Hepa Filtration

Among the different types of active and passive air purification technologies, the latter is based on units with special air filters capable of permanently removing any pollutants.

These filters allow to obtain the right degree of purification based on the sizes of the particles; in fact, the air is forced through a filter, and the impurities are physically captured inside it.

Among the different types of filters, the high efficiency particulate stop filters (HEPA) remove a very high percentage of particles and, especially, the highest classes remove at least 99.97% of 0.3-μm particles (defined by the United States Department of Energy) and are usually more effective at removing larger particles. Following the specifications of the European Union, the filtration capacity of HEPA filters is divided into several classes ranging from >85% to >99.999995%.

HEPA was marketed in the 1950s, and today, this term is used to indicate a highly efficient generic filter [70].

HEPA filters are used in various fields when contamination control is required, such as pharmaceutical production, as well as in hospitals [71].

The composition of the HEPA filters is quite particular, because it is basically a randomly arranged fiber mat made of glass fiber with diameters between 0.5 and 2.0 μm.

Although the air space between the fibers is generally much greater than 0.3 microns, these filtration systems are designed to trap much smaller particles and pollutants, because they remain staked thanks to different mechanisms such as diffusion, interception, inertial impaction, and electrostatic attraction.

In order to improve the level of filtration, HEPA systems are also equipped with prefilters (activated carbon) that remove the coarser impurities (PM10 and pollen particles/10 μm) so that the final HEPA fine filter remains more efficient for several hours, reducing the need to replace or clean it frequently.

In any case, to ensure that a HEPA filter works efficiently, it must be checked and replaced periodically. Not changing a HEPA filter when necessary is a risk, because it could cause, first of all, stress on the device itself and, secondly, an insufficient removal of harmful polluting particles from the air; in addition, attention must be paid to the amount of air passing through the filter, avoiding it bypassing the HEPA filter.

Unlike other air purification systems, HEPA filters do not generate harmful products such as ozone.

Some studies have revealed that it is possible to highlight the growth of microorganisms in bioaerosols in operating rooms even after sterilization, disinfection, and washing due to high humidity, poor ventilation, insufficient disinfection, and floor sweeping. Hence, a HEPA filtration system could help reduce the bioaerosol levels in these environments, including dental environments [72].

## 3. Discussion

The dental literature shows that different dental procedures produce both bioaerosols and droplets that are contaminated with microorganisms mixed with blood, saliva, dental debris, restoration materials, etc. These aerosols represent a real potential pathway for the transmission of diseases between patients and dentists, but there is also the possibility of cross-transmissions between the patients themselves if the decontamination procedures are not ideal. In particular, if we focus our attention on bioaerosols, it seems clear enough that, even if the dental instruments and all surfaces are well-sterilized, an operating room with contaminated air could spread diseases among patients, considering that they spend most of their minutes with their oral cavities open.

Probably the easiest way to eliminate a contaminated bioaerosol is to ventilate the room for several minutes to allow a complete exchange of air, but there is no rapid clinical verification method that certifies that the air present in the operating room is free of contaminants; it is necessary to ensure that all contaminated air has been replaced with fresh and pure air.

Furthermore, simply opening the window may not be sufficient to guarantee the necessary exchange of air; in this case, you should rely on a forced ventilation system.

This method could also be in contrast with maintaining the room temperature at constant and predetermined values, especially in very hot or very cold seasons.

At this point, given the uncertainties of the results and the technical complications that could arise in the implementation of this exchange of air, it would be more appropriate to rely on more scientifically validated systems capable of treating the contaminated air and reducing, if not even zeroing with percentages close to 99%, microbial contaminations.

Now that the risk of spreading COVID-19 is very high, it is necessary to pay particular attention to all the sterilization procedures that should be reviewed, improved, and perhaps used in combinations to obtain a final result that aims to complete the sterilization of all structures present in the operating room, including air, which for some dangerous diseases, such as SARS-CoV-2, is the transmission route.

The latest “no-touch” decontamination technologies could help dentists achieve this important goal.

## 4. Conclusions

In conclusion, continuous efforts are needed to improve traditional manual surface disinfections.

In addition, dentists should consider combining the use of appropriate disinfectants and noncontact decontamination technologies to improve the sterilization of dental operating rooms, especially since the latter methods are independent of the operator.

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
