# Peer review of "Alternative Methods of Sterilization in Dental Practices Against COVID-19"

_ijerph, 2020, doi:10.3390/ijerph17165736_

Round 1

Reviewer 1 Report

The paper is interesting and important. There are a few small points:

  1. Page 2, first line. With "Lethal viruses" I think "Potentially lethal viruses..." is more accurate. 
  2. Page 2, with reference to pathogens in dental practices, and COVID-19 specifically, (around Line 48), please add the reference: Sandle, T. (2020) COVID-19 and dental practice, Dental Nursing, April 2020, pp2-3: https://www.magonlinelibrary.com/doi/full/10.12968/denn.2020.16.4.194
  3. Line 65 on page 2, COV-19 should be written as 'COVID-19'.
  4. Page 5, it will be useful to mention gamma radiation as an example of ionizing radiation. 
  5. Page 11, with HEPA filters it is important to mention that the highest classes of HEPA filter out 99.97% of particles (but not every class of HEPA filter can do this).
  6. The references starts at number '7' rather than number '1', this needs correcting. 

Author Response

Dear reviewer 1,

We appreciated your help in improving our work so we tried to correct it following your suggestions as following:

  1. Page 2, the sentence has been changed as suggested: "Potentially lethal viruses..."
  2. The suggested reference has been added.
  3. COV-19 has been corrected into 'COVID-19'.
  4. Gamma radiation has been mentioned.
  5. We specified that only the highest classes of HEPA filter out 99.97% of particles.
  6. The reference list has been corrected.  

Reviewer 2 Report

Dear authors,

I would like to congratulate you for completing such a relevant and important piece of work amid COVID-19 crisis, in my opinion this paper should be published as soon as possible after some minor corrections and modifications are implemented.

1- Line 58, there is nothing about Materials and methods under heading, so because this is a literature review you do not have to have a Materials and methods heading but if you decide to have this title you need to add your materials and methods. 

2- I do not think that the Table 1 is making any scientific/data contribution to the text of the paper. The contents of the table have already been mentioned in the previous sentence in Lines 78 and 79 and table does not represents any new data/info 

3- Line 178  "others studied have been carried out " is incorrect in terms of English, it should be Other studies 

4- Line 211, 212 need referencing you can use this recent paper  (Barabari P, Moharamzadeh K. Novel Coronavirus (COVID-19) and Dentistry-A Comprehensive Review of Literature. Dent J (Basel). 2020;8(2):E53. Published 2020 May 21. doi:10.3390/dj8020053)

5- line 225,226,227 need referencing 

6- Table 2 is the same as table 1 does not add any specific info and simply illustrates what is written in the text, just get rid of it 

7- Lines 380 till 390 need referencing 

8- Paragraphs starting at lines 418 and 422 should be referenced 

9- Line 439 till 450 need reference 

10- Line 472 to 499 need proper referencing 

11- Line 573 Your reference list start with Reference number 7! the first 6 references have not been included in the list 

Author Response

Dear reviewer 2,

We appreciated your help in improving our work so we tried to correct it following your suggestions as following:

1- We removed Materials and methods because this is a literature review.

2 Table 1 has been eliminated.

3- The sentence "others studied have been carried out " has been corrected.  

4- Line 211, we added the suggested reference (Barabari P, Moharamzadeh K. Novel Coronavirus (COVID-19) and Dentistry-A Comprehensive Review of Literature. Dent J (Basel). 2020;8(2):E53.

5- Proper references have been added.

6- Table 2 has been removed.

7- Proper references have been added.

8- Proper references have been added.

9- Proper references have been added.  

10- Proper references have been added.      

11- The reference list has been corrected.  

Reviewer 3 Report

This is an important and timely paper, which reviews the options for ensuring that, with the current Covid 19 crisis, dental practices remain safe for patients and dental personnel. It will make a helpful contribution to clinical practice, and also understanding.

However, there are some problems that need to be addressed before it can proceed to publication.

First, the early part is not punctuated properly.  It consists of single sentences apparently in paragraphs, with no structure and with thoughts isolated.  As such, it is very difficult to read, though what the article contains is sound and important.  Suddenly, when we get to the Discussion, the sentences are brought together in proper paragraphs, and the whole thing becomes much easier to read.  The authors must revise the whole paper to get it into proper paragraphs.

Second, the second headed Materials and Methods is in appropriately titled.  In a review, keeping to the usual divisions of a scientific paper is not necessary, so this section should be renamed in a way that indicates its content.

In line 47, the word "practises" should be "practices".

In line 228, the word "unbelievable" should be "severe".

References 1-6 are missing and must be included (or the remaining references renumbered...).

Author Response

Dear reviewer 3,

We appreciated your help in improving our work so we tried to correct it following your suggestions as following:

-We have tried to change the first part of our paper in order to make it more readable.

-We removed Materials and methods because this is a literature review.

-In line 47, the word "practises" has been corrected.  

-In line 228, the word "unbelievable" has been changed in to "severe".

-The reference list has been corrected.  

Round 2

Reviewer 3 Report

This paper has been improved slightly, but it still makes the mistake of being mainly comprised of single sentence "paragraphs" for much of the article.  Sentences MUST be collected together into paragraphs that explore a single idea or concept.  This is how English works, and it gives the reader clues about how to read the words in a way that promotes understanding.  The authors (or someone) must correct this before the paper can be published.

Author Response

Dear reviewer, thank you for your help and for the useful suggestions to improve our scientific work. We tried to improve the readability and fluency of the entire text in the hope that it will be published.

Round 3

Reviewer 3 Report

My main concern had been with the way the earlier version was set out, without proper paragraphs.  This has now been largely addressed by the authors.  the content remains scientifically sound and the whole piece makes an important contribution that is timely during the international crisis caused by Covid-19.  I am now happy for this article to proceed to publication.